# Mitigating the Recrystallization of a Cold-Worked Cu-Al_2_O_3_ Nanocomposite via Enhanced Zener Drag by Nanocrstalline Cu-Oxide Particles

**DOI:** 10.3390/nano13192727

**Published:** 2023-10-08

**Authors:** Ramasis Goswami, Alex Moser, Chandra S. Pande

**Affiliations:** 1Materials Science and Technology Division, Naval Research Laboratory, Washington, DC 20375, USA; alex.moser@nrl.navy.mil; 2Naval Research Laboratory, Washington, DC 20375, USA; chandrasp22@gmail.com

**Keywords:** microstructure, interfaces, recrystallization, nanosized oxide dispersion, TEM

## Abstract

The strength of metals and alloys at elevated temperatures typically decreases due to the recovery, recrystallization, grain growth, and growth of second-phase particles. We report here a cold-worked Cu-Al_2_O_3_ composite did not recrystallize up to a temperature of 0.83T_m_ of Cu. The composite was manufactured through the internal oxidation process of dilute Cu-0.15 wt.% Al alloy and was characterized by transmission electron microscopy to study the nature of oxide precipitates. As a result of internal oxidation, a small volume fraction (1%) of Al_2_O_3_ particles forms. In addition, a high density of extremely fine (2–5 nm) Cu_2_O particles has been observed to form epitaxially within the elongated Cu grains. These finely dispersed second-phase Cu_2_O particles enhance the Zener drag significantly by three orders of magnitude as compared to Al_2_O_3_ particles and retain their original size and spacing at elevated temperatures. This limits the grain boundary migration and the nucleation of defect-free regions of different orientations and inhibits the recrystallization process at elevated temperatures. In addition, due to the limited grain boundary migration, a bundle of stacking faults appears instead of annealing twins. This investigation has led to a better understanding of how to prevent the recrystallization process of heavily deformed metallic material containing oxide particles.

## 1. Introduction

Copper alloys are useful candidates for elevated temperatures and high heat flux (100 MW m^2^) applications, such as heat sinks, rocket engine combustion chambers, nozzle liners, and reusable launch vehicle (RLV) technologies, where a high thermal conductivity, good elevated temperature strength, resistance to creep, and low cycle fatigue are needed [1,2,3,4]. Several such Cu alloys, including Cu-Ag-Zr [1,5], Cu-Zr (AMZIRC) [1,4], Cu-Al_2_O_3_ (GlidCop) [1,6], Cu-Cr-Nb (GRCop) [7,8], and Cu-Cr-Zr [9,10], have been developed, which exhibit high thermal and electrical conductivities and strength at elevated temperatures. For example, the alloy AMZIRC has an ultimate tensile strength (UTS) of 500 MPa at room temperature, and its strength decreases to 260 MPa at 500 °C and then drops significantly below 50 MPa when above 600 °C [1]. The decrease in strength at elevated temperatures is mostly due to the recrystallization, grain growth, and growth of second-phase particles or precipitates. The Cr-rich GRCop-84, developed for high heat flux applications, shows good elevated temperature strength, which stems from the precipitation hardening of Cr_2_Nb precipitates. These precipitates reduce grain growth by pinning grain boundaries at elevated temperatures [1]. Additionally, copper-based metal matrix composites find applications in the electronics and automotive industries due to their excellent wear resistance, corrosion resistance, mechanical properties, and electrical properties. They are also promising materials in future fusion reactors. This is due to their high thermal conductivity and good mechanical properties.

However, one of the critical drawbacks of these composites is the rapid loss of plastic strength at elevated temperatures close to the melting point, owing to rapid recrystallization and grain growth. Controlling recrystallization and grain growth is, therefore, very important to retain strength at elevated temperatures. We report here a heavily cold-worked Cu-Al_2_O_3_ composite that did not recrystallize up to a temperature of 0.85T_m_ of Cu. It has been reported in GlidCop (Cu-Al_2_O_3_) [1,6,10] that the recrystallization process of the Cu matrix can be inhibited at relatively elevated temperatures with a dispersion of a small amount of Al_2_O_3_ precipitates (~1%). A number of boundaries might not be effectively pinned by such a small volume fraction of Al_2_O_3_ particles as the Zener pinning depends on the fraction and the size of precipitates. One could achieve a higher level of Zener pinning if the particle size is significantly small, even with the lower volume fraction (1%) of the precipitate. The objective of this research is to investigate fine-scale microstructure using transmission electron microscopy (TEM) and to revisit the mechanism of mitigation of recrystallization of internally oxidized cold-rolled Cu at relatively elevated temperatures. The Cu-Al_2_O_3_ composite was manufactured through the internal oxidation process of dilute Cu-0.15 wt.% Al alloy.

## 2. Experimental Section

Several powder metallurgical methods, such as ball milling of powder mixtures and internal oxidation of alloy powders, can be implemented to incorporate Al_2_O_3_ nanoparticles into the Cu matrix in the solid state as opposed to conventional techniques, such as melting and casting, which result in segregation in the melt due to differences in density between Cu and ceramic particulates. In this case, initially, the Cu-dilute Al powders are made initially by melting the alloy and then using spray forming in an inert atmosphere. For internal oxidation, the alloy powders were then heated to an elevated temperature in a controlled oxidizing atmosphere. The powders were then consolidated at elevated temperatures and cold rolled to form plates. We used a commercial cold-rolled plate of GlidCop Al-15 to investigate the fine-scale microstructure. It contains 0.15 wt.% Al and 0.17 wt.% O. The volume fraction of Al_2_O_3_ particle is ~1% in as-received commercial GlidCop Al-15 [10]. The oxygen level was determined by the inert gas fusion-infrared absorption method. The electrical and thermal conductivity of the internally oxidized Cu was observed to be ~6–8% lower as compared to pure Cu in the as-received condition [10]. The rolled plates were heated to 900 °C and then cooled in air to room temperature. The strain level of the as-received rolled plates of GlidCop Al-15 was ~80% [10]. The samples were then characterized by X-ray diffraction (XRD) using a Rigaku 18 kW X-ray generator (Rigaku Corporation, Tokyo, Japan) and a high-resolution powder diffractometer utilizing Cu-Kα_1_. For TEM, samples in the as-received and in the annealed conditions were prepared by initially polishing the disk samples mechanically and finally by thinning them in an ion mill with a gun voltage of 4 kV and a sputtering angle of 10°. A JEOL 2200 analytical transmission electron microscope (JEOL USA Inc., Peabody, MA, USA) operating at 200 keV was used to characterize the microstructure and interfacial characteristics of the oxide particulates. To minimize the strong strain contrast due to cold rolling, we use multibeam imaging to reveal the grain boundaries and coarse precipitate particles. To study the structure of oxide particles formed during the oxidation process, we performed high-resolution TEM (HRTEM) imaging. 

## 3. Results

The in situ internal oxidation process of Cu-Al alloy results in the formation of Al_2_O_3_ within grains as well as at grain boundaries. In this composite, the estimated volume fraction of Al_2_O_3_ is around 1% [10,11,12]. Note that the Al and oxygen content in the composite is 0.15 and 0.17 wt.%, respectively. The Al atoms in the Cu matrix become oxidized to form Al_2_O_3_ due to the reaction, 4(Cu-Al) + 3O_2_ = 4Cu + 2Al_2_O_3._ After the internal oxidation process, the powders were consolidated at elevated temperatures and cold rolled to form plates. The bright-field TEM image (see Figure 1a) of the as-received cold-rolled plate of GlidCop Al-15, obtained along the rolling direction, shows elongated columnar grains as a result of cold working. The average width of the columnar grains is ~100 nm, measured from a number of TEM images obtained from different areas of the sample. We utilized TEM images for this measurement as the XRD peak broadening is influenced by the higher dislocation density. In addition, we observe alumina particles at grain boundaries (see the inset of Figure 1a). These elongated Cu grains are highly oriented with a very strong 220 reflection of Cu, while the 111 peak is very weak (see Figure 1b). It is known that cold working on Cu produces a brass-type texture, which can lead to a strong 220-type reflection. Cu is known to form brass-type texture upon heavy deformation (cold working) with very strong 220 [13,14]. The major slip system in copper is {111} <110>. During the rolling process, the majority of {220} planes tend to orient towards the rolling plane. As the {111} <110> slip system becomes active, it aligns the {110} planes parallel to the plane containing the rolling directions [14]. To minimize the strain contrast as a result of cold rolling, we use multibeam imaging to clearly show the elongated grains and oxide precipitates within grains and grain boundaries. In Figure 2a, the multibeam image shows the elongated grains with fine dispersion oxide particles within a grain. These oxide particles pin the dislocations (see Figure 2b) and increase the yield strength by the Orowan mechanism. 

To study the nature of the aluminum oxide particle formed during the oxidation process, we performed HRTEM imaging. These studies show that the aluminum oxide particles are not distributed uniformly in the matrix, and the particle size within the matrix grain ranges from 10 to 20 nm (see Figure 3a). The HRTEM image (see Figure 3b), close to the 110 zone, shows gamma alumina (γ-Al_2_O_3_) with {111} lattice spacing of ≈4.6 Å, consistent with the {111} lattice spacing of known γ-Al_2_O_3_ with lattice parameter a ≈7.97 Å. The corresponding FFT is shown in Figure 3c. One can observe that it is cube-on-cube orientated with the matrix. In some cases, we observe that the orientation of γ-Al_2_O_3_ has deviated from the cube-on-cube orientation (see Figure 3d). At grain boundaries, however, we observe coarse alumina particles, mostly faceted and parallel to the boundary. In some cases, the size (length) is in the range of 200 to 300 nm (see Figure 4a). A number of grain boundaries were observed to be free of these oxide particles. Figure 4b is an HRTEM image obtained from the area indicated as rectangular box “A” in Figure 4a, showing the interface and a portion of the faceted Al_2_O_3_ particle. The corresponding FFT shows it is a γ-Al_2_O_3_ particle close to the zone 112 at the grain boundary. 

The cold-worked samples were heated to 900 °C and then cooled to room temperature. TEM observations show that the grains (G-1 through G-8) were still columnar (see Figure 5a) even after exposure to elevated temperatures, suggesting the recrystallization did not occur at high homologous temperatures. We examined different areas of the sample and did not observe nucleation of equiaxed grains and dislocation cells or subcells within the elongated grains (see inset of Figure 5a). Note that the recrystallization process usually produces equiaxed grains that are substantially free of dislocations. Instead, due to thermal exposure, we observe a change in grain orientation as evidenced by the increase in the intensity of the 111 reflection at the expense of the intensity of the 220 reflection (see Figure 5b). In fact, the intensity ratio of 220/111 decreases from 17.5 to 1.33 (see Figure 1b and Figure 5b for comparison) after annealing. In addition, we observe limited grain coarsening in some areas using TEM. We, in fact, examined a number of areas (around 10 to 15 instances) and observed localized grain coarsening. Figure 6a is a multibeam image showing the coarsening of grains A and B at the expense of grain C. At certain locations, the width of the grain has increased significantly to 400 nm from the average grain width (Figure 6a) of 100 nm. This indicates that the grain coarsening is related to the growth of 111-oriented grains at the expense of some of the 220-oriented grains. Note that the intensity of the 220 peak is still greater than that of the 111 peak after annealing. However, the grain coarsening does not induce any recrystallization in this composite. Upon heating, XRD shows that the intensity of the 111 peak increases at the expense of the 220 peak (see Figure 1b and Figure 5b). Note the significant change in the intensity ratio of 220/111 upon heating to elevated temperature as compared to the intensity ratio of 220/111 in the as-received condition. The increase in intensity of the 111 peak is mostly related to the growth of 111-oriented grains, which grow at the expense of the 220-oriented grains upon heating. This reduces the overall energy of the system. Note that we have a small number of grains with 111 orientations in the as-received condition. As we have not observed any recrystallization, the reorientation of grains upon heating is the result of the localized growth of the 111-oriented grains. In addition, in some areas, we observed that the grain boundaries were pinned by Al_2_O_3_ particles (see Figure 6b,c). Note that the volume fraction of Al_2_O3 particles is just ~1%, and a significant fraction of grain boundaries are not pinned by these particles.

Most of the dislocation is still pinned even after heating (see inset of Figure 5a), although some amount of dislocation redistribution has been expected to take place after heating to a high homologous temperature. In fact, we observe the low-angle grain boundary formation in some areas as a result of recovery or redistribution of dislocations. Figure 7a is a low-magnification bright-field image showing the distribution of dislocations and a low-angle grain boundary. The dislocations are shown by arrows. A higher magnification image of the low-angle boundary, along with the oxide particles, obtained from the square area indicated in Figure 7a, is shown in Figure 7b. Figure 8 is the HRTEM image of this low-angle boundary, showing the periodic dislocations and a small tilt of the 111 plane across the boundary, demonstrating some amount of recovery taking place during annealing. The angle of the low-angle grain boundary is approximately 10^0^.

Additionally, we observe annealing twins in elongated grains oriented close to the 110 zones in this composite. Annealing twinning in pure Cu usually occurs during the growth of the recrystallized grains in the deformed matrix. Note, here, that no recrystallization has been observed in the present case in the deformed Cu-Al_2_O_3_ matrix. Figure 9a is a low-magnification image showing twins in an elongated grain, running through the grain and ending at the other side of the grain boundary. Steps associated with twins at either side of grain boundaries are observed (see Figure 9b). The high-resolution TEM (HRTEM) image (see Figure 9c), close to the 110 zone, shows twins consisting of bundles of stacking faults in 111 lattice planes in the copper matrix. The fault packets are shown. In some cases, these faults are initiated and ended within the grain (see Figure 9c). The fast Fourier transform (FFT) from the stacking fault bundle shows twin spots (see the inset) and streaks along the 111 direction.

The internal oxidation process can also result in the formation of copper oxide particles, a small fraction of dissolved oxygen, and copper–aluminum–oxygen clusters in the matrix. Such a fine dispersion of oxide particles and clusters increases the hardness and elastic modulus of the composite [11] as compared to commercially pure Cu. In fact, we observed a very fine dispersion of Cu-oxide particles within the elongated grains of Cu before and after the annealing treatment using HRTEM. Figure 10a shows the dispersion of cubic Cu_2_O particles in the as-received condition, as shown by arrows. The corresponding FFT (see the inset) shows Cu_2_O reflections in a cube-on-cube orientation with the Cu matrix. The d-spacing, 0.246 nm, conforms to 111 Cu_2_O. We extracted an inverse FFT (IFFT) image (see Figure 10b) from the 111 Cu_2_O spots to show these oxide particles indicated by circles. The particle size ranges from 2 to 4 nm in diameter. One could estimate the amount of Cu_2_O in the matrix from a number of HRTEM-IFFT images, which turned out to be ~15%. The chemical composition can, in principle, be obtained by energy dispersive spectroscopy (EDS) or EDS mapping. However, the size and spacing of the Cu-oxide particles are extremely small, which could result in significant spurious counts from the Cu matrix. In addition, the EDS information will not be quantitative as they are oxide particles. Thus, lattice imaging would be the efficient method to obtain the stoichiometry of these nanosized Cu-oxide particles. We looked at a number of regions/grains to make sure that we had enough statistics. Upon exposure to elevated temperature, we observed that these Cu_2_O particles did not grow significantly (see Figure 11a,b). They retain their original size and spacing as these oxide particles originally form at high temperatures during the internal oxidation process of Cu-Al powders.

## 4. Discussion

In the Cu-Al_2_O_3_ composite material under investigation, the recrystallization process does not take place even at temperatures > 0.8T_m_, as evidenced by the multibeam imaging (see Figure 5 and Figure 6). In pure Cu [15], on the other hand, the recrystallization process readily occurs in the deformed matrix, and annealing twins form mostly during the growth stage after recrystallization. It has also been reported that these annealing twins play an important role in the nucleation of recrystallized grains. It is known that fine-scale precipitates will slow down the recrystallization process. For example, in the Al-Sc system [16], the fine-scale Al_3_Sc precipitates increase the Zener pinning effect at elevated temperatures. However, at higher temperatures, these precipitates were observed to grow, the Zener effect was minimized/reduced, and recrystallization occurred. In our system, we find that the nanocrystalline semicoherent Cu_2_O precipitates within grains in addition to Al_2_O_3_ particles (see Figure 10). These particles do not grow significantly even after annealing close to the melting temperature of Cu (see Figure 11). These Cu_2_O precipitates exert significantly higher Zener drag force as compared to Al_2_O_3_ particles. 

As is well known, the driving force for recrystallization and grain growth in cold-worked metal is mostly the stored deformation energy in the form of dislocations. The driving force is Δ*P* = 0.5 *ρµb*^2^, where *ρ* is the density of dislocations and *µb^2^* is the energy of dislocations. The *µ* is the shear modulus, and “*b*” is the Burgers vector. The as-received sample is heavily deformed (the deformation level is 80%), suggesting the driving force for recrystallization is rather strong. The presence of strain concentration (hot spot) due to rolling would increase the tendency to recrystallize. As we have not observed any recrystallization, it suggests the strain concentration might be less. When considering the density of dislocation as 10^15^ m^−2^ for heavily cold-worked alloy and the dislocation energy as 10^−8^ Jm^−1^, the estimated driving force is 10 MPa. However, the driving force, 2γ/r, for grain boundary migration due to grain boundary energy would be two to three orders of magnitude smaller, as the radius, r, of the elongated grain (longer side) would be much higher. Thus, the recrystallization would be mostly driven by the stored energy. Although the segregation of oxygen at grain boundaries can alter the grain boundary energy, the driving force contribution for grain boundary migration would be much smaller as compared to the stored energy. 

We have observed limited grain growth (thickening) and no recrystallization for the sample heated to elevated temperatures (900 °C). Some of the elongated grains grow by increasing their width to some extent and increase the population of 111-oriented grains, as the intensity of 111 increases at the expense of 220. The 111-oriented grains will have lower energy compared to the 220-oriented grains. The grain boundary also experiences a drag effect due to the presence of Al_2_O_3_ particles within the grains and at grain boundaries. Drag develops as a result of the attractive force between the particle and grain boundary, as it reduces the part of the grain boundary upon contact with the particle, which is known as the Zener force. The Zener force is proportional to the number of particles at grain boundaries or the volume fraction of particles in the system. The reduction of grain boundary free energy per particle, ΔG, and the corresponding Zener force, Z_f_, are given below [17]:ΔG = γ (1 − πr^2^)(1)
Z_f_ = πrγ(2)
where r is the radius of the particle, and γ is the interfacial energy of the precipitate matrix interface. Note, here, that the volume fraction of Al_2_O_3_ particles is ~1%, suggesting the effect due to the Zener force would be small. Therefore, the contribution of the Zener effect on recrystallization behavior would be small. In a number of cases, we observed the portion of the grain boundary is free of Al_2_O_3_ particles. 

Thus, it is reasonable to discuss another alternative mechanism in this system. TEM results show, in addition to Al_2_O_3_ particles, the presence of nanocrystalline (2 to 5 nm in diameter) semicoherent Cu_2_O particles within the matrix grains (see Figure 10 and Figure 11) in as-received and annealed conditions, respectively. In addition to Al_2_O_3_ particles, one could expect copper oxide, a small fraction of oxygen (the solubility of oxygen in Cu at the eutectic temperature is 0.008 wt.%), to be in the matrix during the internal oxidation process. Such a fine dispersion of oxide particles increases strength and could inhibit the recrystallization of the copper matrix at temperatures close to the melting point of copper. These Cu_2_O particles are oriented in a cube-on-cube relation within the matrix grain. The grain boundary migration will result in an orientation change in the particle. Consequently, the interfacial energy change is given by [17]
ΔG = 4πr^2^ (γ_2_ − γ_1_)(3)
where γ_1_ and γ_2_ are the interfacial free energies of semicoherent Cu_2_O particles in the growing and vanishing grains, respectively. The experimentally observed volume fraction of Cu_2_O particles using HRTEM is ~15%, which is considerably higher than the volume fraction of Al_2_O_3_ particles, suggesting substantial Zener drag due to Cu_2_O particles. The Zener drag depends on the fraction and the size of precipitates. A higher level of Zener pinning as the Cu_2_O particle size is significantly smaller in the range of 2 to 4 nm in diameter. The recrystallization nucleation, i.e., the nucleation of defect-free regions, requires the presence of supercritical nuclei of different orientations. The nucleation of deformed Cu grain recrystallization does not occur by thermal fluctuations. Usually, a subgrain or cell structure greater than the critical size present in the deformed system can act as a possible nucleation center. The critical size is given by [17]
*r_c_* = 2 γ/Δ*P*(4)

The estimated critical size, *r_c_*, is around 100 nm, considering the grain boundary energy to be 1.0 Jm^−2^ and Δ*P* to be 10 MPa. Usually, a subgrain or cell structure greater than the critical size present in the deformed system can act as a possible nucleation center. TEM results show no obvious dislocation cell formation after annealing, suggesting recrystallization nucleation events may not occur. In addition, as the Cu_2_O particles are cube-on-cube oriented in the deformed matrix, the formation and the growth of the recrystallized grains of different orientations would be energetically expansive and produce higher Zener drag, according to Equation (3), due to the difference in the interfacial free energies of semicoherent of Cu_2_O particles in the growing and vanishing grains. 

Additionally, the migration of the grain boundary of the deformed grain is required for the formation of viable recrystallization nuclei and annealing twins. In this case, the grain boundary migration is restricted due to the Zener drag exerted by the second-phase particles. Here, we estimate the Zener drag due to the presence of the nanocrystalline Cu_2_O and Al_2_O_3_ particles. The particles in contact with the grain boundary per 1 m^2^ of a boundary are n = 2rN, where r is the radius of the particle and N is the number of particles per unit volume [17]. The volume fraction of the particle, V_P_, is (4/3) πr^3^N. Considering the experimentally estimated volume fraction of Cu_2_O particles to be 15%, the number of particles, n, in contact with the boundary is 16 × 10^15^ per m^2^ of the boundary. The maximum pinning Zener drag force [17] is Z_F_ = (3/2) V_p_γ_p_/r, where γ_p_ is the particle/matrix interfacial energy. The estimated maximum pinning force exerted by the Cu_2_O particles is 5 × 10^7^ Jm^−3^, considering the γ_p_ to be 0.5 Jm^−2^. Similarly, the estimated maximum Zener force exerted by Al_2_O_3_ particles is 7.5 × 10^4^ Jm^−3^, indicating the pinning force exerted by Cu_2_O particles is three orders of magnitude greater than that of Al_2_O_3_ particles. This limits the migration of the boundary consistent with the experimental observations of the limited grain growth and the fault bundle formation instead of annealing twins. Note that the volume fraction of Al_2_O_3_ particles in the composite is ~1%, and the particle size is considerably larger (see Figure 4) as compared to Cu_2_O. 

Twinning is a mode of plastic deformation in addition to slip. Two types of twins are known as deformation twins and annealing twins. Annealing twins are more likely to form when grain boundaries migrate as pointed out by Pande et al. [18,19]. In 1963, Dash and Brown [20] made the first observations on the early stages of annealing twins. They also showed, for the first time, that twin nuclei consist of stacking fault packets having complicated morphology. Their explanation for the formation of the observed features lacked mechanistic details. A microscopic model based on nucleation of partial dislocations for the formation of annealing twins in fcc crystals was proposed by Mahajan et al. [21]. In their paper, it is argued that Shockley partial loops nucleate on consecutive {111} planes on a moving boundary by growth accidents occurring on migrating {111} steps associated with the boundary. The higher the velocity of the boundary, the higher the twin density. In the present case, we, too, observe early-stage twin formation as the twins are not fully developed due to limited grain boundary migration. Here, the twins are not sharp and are associated with bundles of stacking faults (see Figure 9). Usually, twinning proceeds from a moving high-angle grain boundary. As stated earlier, the driving force, which is the stored energy, is high in this case as it is heavily deformed. Although the presence of twinning has been reported to be associated with the low driving force for recrystallization [13], in this case, the fault bundles are obviously not due to the low driving force. Our results suggest a bundle of faults on 111 lattice planes formed due to the passage of partials as a result of boundary migration. In this case, the fault bundles did not collapse to form sharp twins or twin chains due to limited grain boundary migration.

## 5. Summary and Conclusions

In summary, we report here the absence of recrystallization of cold-worked Cu-Al_2_O_3_ composites, which were heated to 0.85T_m_ of Cu, and the possible mechanisms of this absence. The composite was manufactured through the internal oxidation of dilute Cu-0.15 wt.% Al alloy. We investigated the fine-scale microstructure and interfacial characteristics of the Cu(Al)-Al_2_O_3_ composite processed in the solid state using TEM and demonstrated the internal oxidation of Cu(Al) produces a fine dispersion of oxide particles and clusters in the Cu matrix. TEM studies show no recrystallization of the elongated Cu grains. Using HRTEM, we observe a high density of extremely fine (2–5 nm) cubic Cu_2_O particles formed epitaxially within the Cu matrix, in addition to Al_2_O_3_ particles. The following conclusions can be made.

Internal oxidation of a copper-aluminum matrix produces a fine dispersion of metastable γ-Al_2_O_3_, Cu_2_O, and nanoclusters of Cu-O in a Cu matrix. The nanoparticles of Cu_2_O range from 2 to 5 nm. A cube-on-cube orientation relationship has been observed between the copper matrix and the Cu_2_O. The estimated volume fraction of Cu_2_O is 15%, which is significantly higher than Al_2_O_3_;Cold working produces elongated grains with brass texture, and the XRD studies showed that the intensity ratio of 220 to 111 is high in the as-received condition. Upon exposure to elevated temperature, the intensity ratio of 220/111 peak intensity changes significantly as a result of grain coarsening of 111-oriented grains at the expense of 220-oriented grains;The presence of oriented Cu_2_O nano-dispersion throughout the Cu matrix inhibits the grain boundary migration due to Zener drag and recrystallization at high homologous temperatures. The Zener drag force is three orders of magnitude higher due to the presence of nanocrystalline Cu_2_O particles as compared to Al_2_O_3_ particles;Recrystallization nucleation events do not occur as a cell structure greater than the critical size of 100 nm is needed for a possible nucleation center. TEM results show no obvious dislocation cell formation after annealing, as most of the dislocations are pinned by the oxide particles;The early stage of twin formation in the form of a bundle of faults, as the twins are not fully developed, is due to the limited migration of the grain boundary.

## Figures and Tables

**Figure 1 nanomaterials-13-02727-f001:**
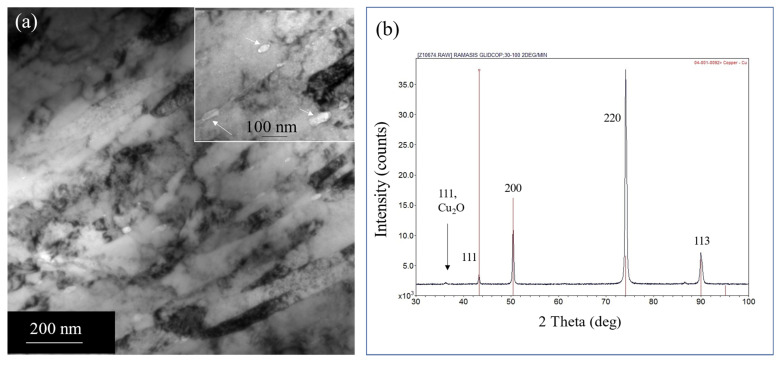
(**a**) TEM image showing the elongated grains of a cold-worked copper–alumina composite. The alumina particles at grain boundaries are shown by arrows as an inset (**b**) XRD of Cu-Al_2_O_3_ composite after cold work, showing the Cu peaks with a strong 220 peak and a very weak 111 peak. The powder diffraction lines of Cu were superimposed with the data for comparison. The red lines represent the powder diffraction lines of Cu, and the X-ray diffraction peaks from the sample are presented with black.

**Figure 2 nanomaterials-13-02727-f002:**
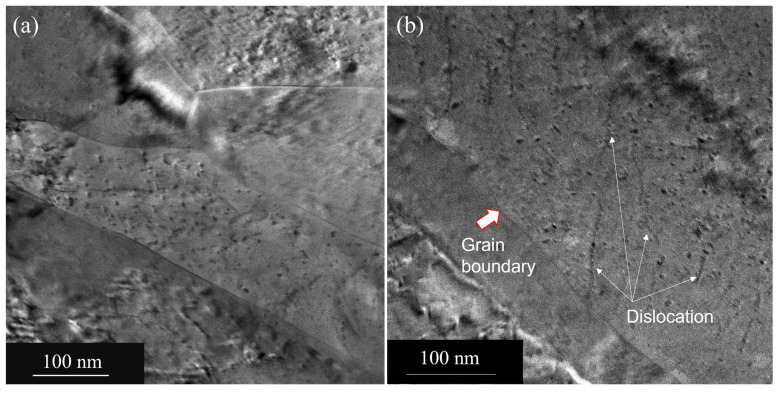
(**a**) A multibeam TEM image showing fine oxide particles in the elongated grains of the matrix. (**b**) A multibeam image showing the pinning of dislocations by these particles in the matrix grain. Dislocations are shown by arrows.

**Figure 3 nanomaterials-13-02727-f003:**
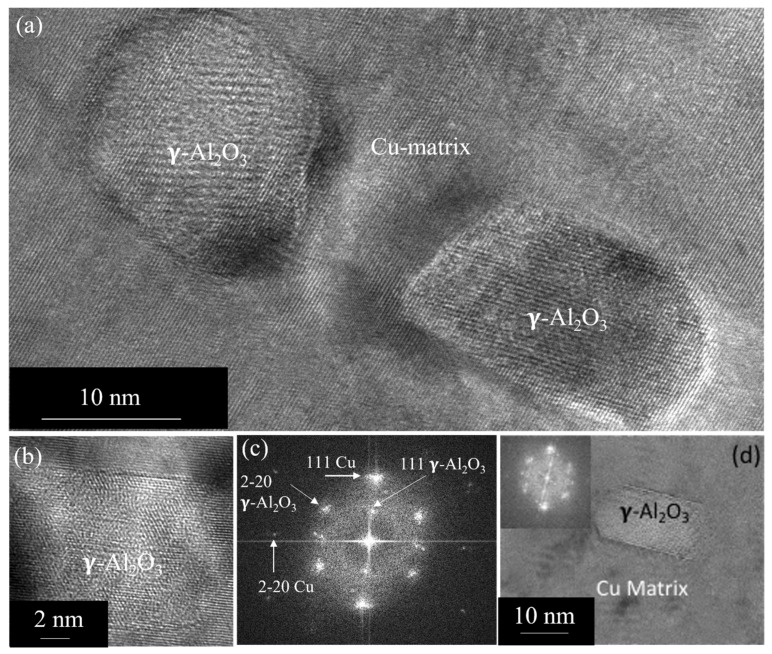
(**a**) HRTEM image showing γ-Al_2_O_3_ particles along the 011 zone showing strain contrast. (**b**) HRTEM showing a γ-Al_2_O_3_ particle close to cube-on-cube orientation relation with the matrix. (**c**) Corresponding FFT from the particle and the matrix. (**d**) HRTEM image showing γ-Al_2_O_3_ particles deviated from the cube-on-cube relation.

**Figure 4 nanomaterials-13-02727-f004:**
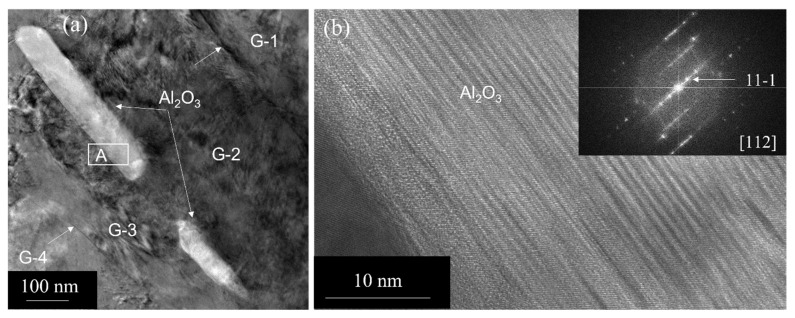
(**a**) A multibeam TEM image showing γ-Al_2_O_3_ particles at grain boundaries. Grains are indicated as G-1 through G-4, and boundaries are shown by arrows. (**b**) HRTEM image showing γ-Al_2_O_3_ particle along the 112 zone.

**Figure 5 nanomaterials-13-02727-f005:**
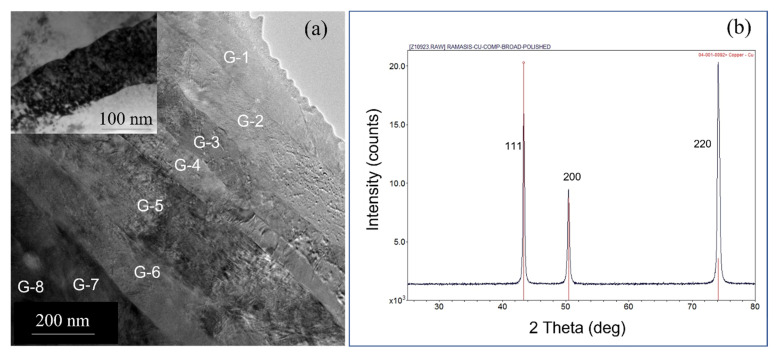
(**a**) A multibeam TEM image showing the elongated grains (G-1 through G-8) after annealing at 900 °C. Note that no recrystallization has been observed. Inset (a bright-field image close to two beam conditions) shows the dislocation distribution in a grain. (**b**) XRD of the Cu-Al_2_O_3_ composite after annealing. Note the change in 220/111 intensity ratio as compared to the as-received cold-worked sample. The powder diffraction lines of Cu were superimposed with the data for comparison. The red lines represent the powder diffraction lines of Cu and the diffraction peaks from the sample are presented in black.

**Figure 6 nanomaterials-13-02727-f006:**
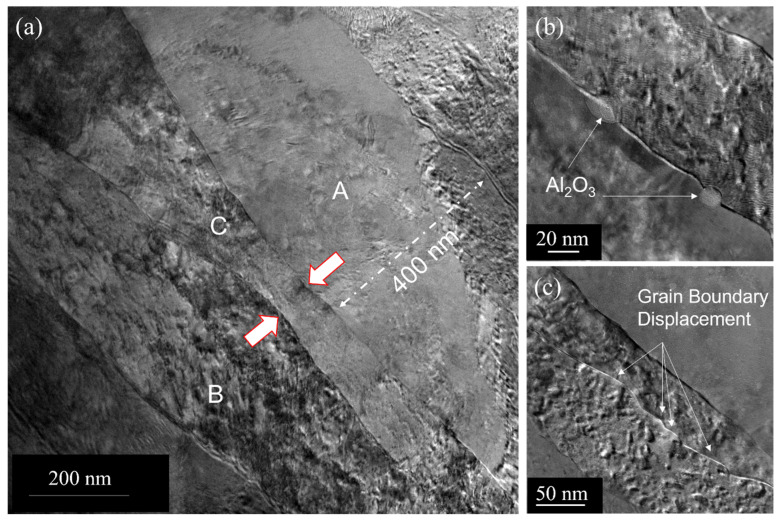
(**a**) TEM image showing grain coarsening in some areas due to annealing at 900 °C. One can see the coarsening of elongated grains A and B at the expense of grain C. Thick arrows indicate the direction of grain boundary migration. (**b**) A bright-filed TEM image showing the pinning grain boundary by Al_2_O_3_ particles indicated by white arrows. (**c**) A bright-filed TEM image showing the pinning grain boundary and displacement by oxide particles (Cu_2_O and Al_2_O_3_) as shown by white arrows.

**Figure 7 nanomaterials-13-02727-f007:**
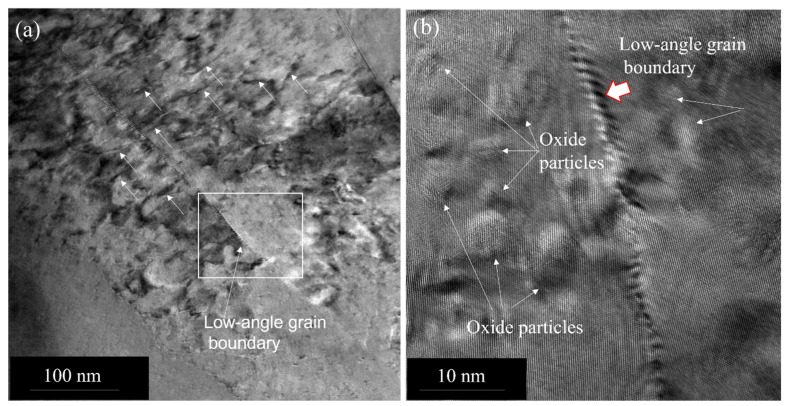
(**a**) TEM image showing the dislocation rearrangements indicated by arrows and a low-angle grain boundary formation. (**b**) A higher magnification image showing the low-angle boundary with a thick arrow. The oxide particles are shown by arrows.

**Figure 8 nanomaterials-13-02727-f008:**
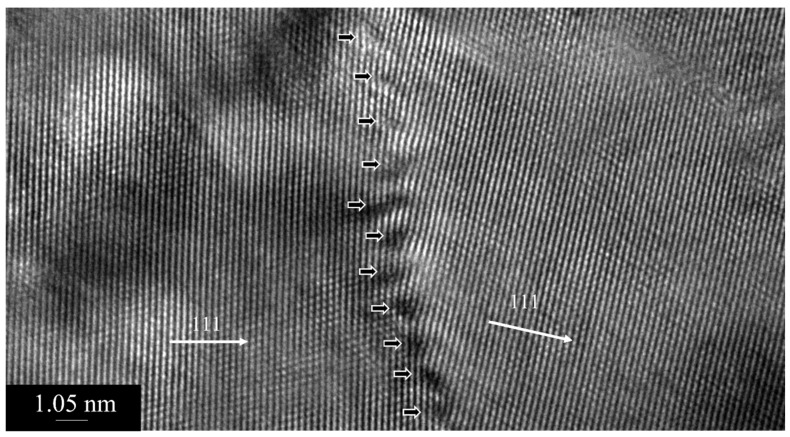
HRTEM image of the low-angle boundary, as shown in Figure 7b, showing periodic dislocations with thick arrows. The 111 direction across the boundary is shown by arrows.

**Figure 9 nanomaterials-13-02727-f009:**
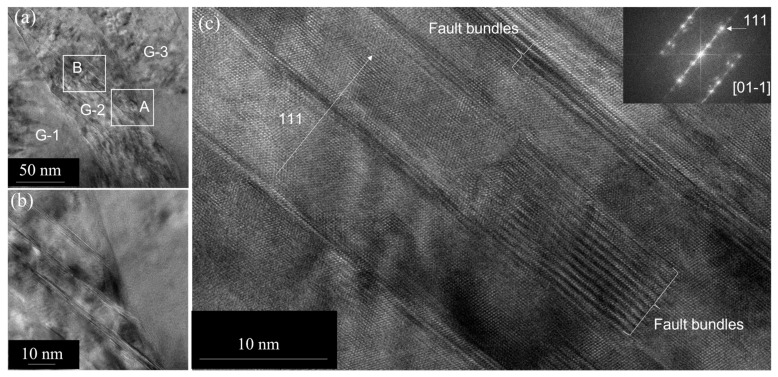
(**a**) A low-magnification TEM image showing an elongated grain with a high density of faults upon annealing. (**b**) HRTEM image close to the 110 zone showing microtwins in the Cu matrix obtained from square area A in (**a**). (**c**) A high-magnification HRTEM showing that it consists of a bundle of faults obtained from square area B in (**a**). The corresponding FFT showing the twin-related reflections is given as an inset.

**Figure 10 nanomaterials-13-02727-f010:**
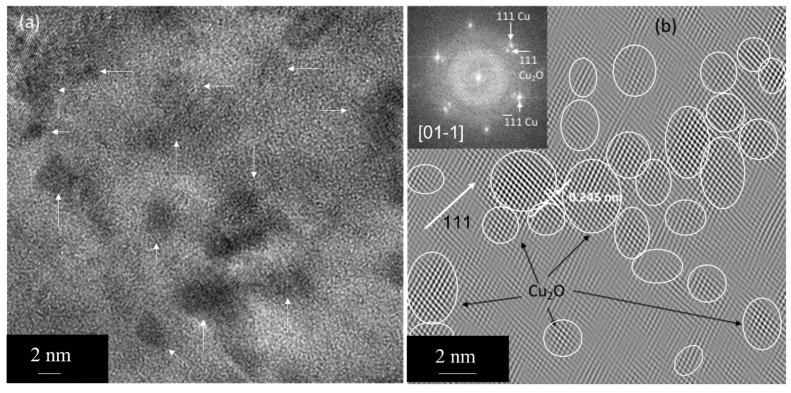
(**a**) HRTEM image showing the dispersion of copper oxide (Cu_2_O) particles, as shown by white arrows, in Cu matrix in as-received condition. (**b**) IFFT image showing nanocrystalline Cu_2_O particles indicated by circles as shown by black arrows. The 111 direction in (**b**) is indicated by a white arrow. The FFT is given as an inset.

**Figure 11 nanomaterials-13-02727-f011:**
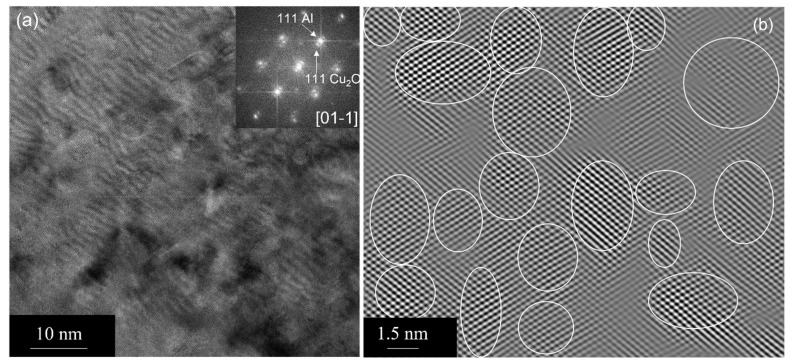
(**a**) HRTEM image showing the dispersion of copper oxide (Cu_2_O) particles upon annealing in Cu matrix. (**b**) IFFT image showing nanocrystalline Cu_2_O particles indicated by circles. The FFT is given as an inset.

## Data Availability

Here data is given in form of TEM images.

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
