# Peer review of "Mitigating the Recrystallization of a Cold-Worked Cu-Al2O3 Nanocomposite via Enhanced Zener Drag by Nanocrstalline Cu-Oxide Particles"

_nanomaterials, 2023, doi:10.3390/nano13192727_

Round 1

Reviewer 1 Report

The authors Goswami et al. have investigated the strategy for mitigating the recrystallization of Cu-Oxide particles. The results are interesting. However, some minor issues need to be addressed before publishing.

1.    In the experimental part, the supplier, impurities, and other information of all samples are required to be supplied.

2.    The method for measuring the concentration of Al and O needs to be mentioned.

3.    The resolution of Fig. 1 is too low.

4.    In Figs, 1-11, the format (background) of scale bars are not uniform.

Author Response

Reviewer-1

The authors Goswami et al. have investigated the strategy for mitigating the recrystallization of Cu-Oxide particles. The results are interesting. However, some minor issues need to be addressed before publishing.

  1. In the experimental part, the supplier, impurities, and other information of all samples are required to be supplied.

Ans: We have modified the information in the revised text.

  1. The method for measuring the concentration of Al and O needs to be mentioned.

Ans: The commercial alloy contains 0.15 wt.% Al and 0.17 wt % O.  Around 0.15 wt% Al was added to make a dilute Cu-Al alloy before oxidation. The oxygen level was determined by the inert gas fusion-infrared absorption method. The electrical and thermal conductivity of the internally oxidized Cu was observed to be ~ 6-8% lower as compared to pure Cu in the as-received condition. This is specified in the product in the as-received condition by the company.

The resolution of Fig. 1 is too low.

Ans: We have modified the resolution (JPEG image) of Fig. 1.

  1. In Figs, 1-11, the format (background) of scale bars are not uniform.

We modified the background of all figures in the revised texts. 

Reviewer 2 Report

Review of the article.

The article expresses the reasons for the lack of recrystallization of the Cu-Al. The results are interesting, but several questions arise.

At what time and at what oxygen pressure does the internal oxidation of the alloy components and the formation of Al2O3, Cu2O.

What conditions need to be created to prevent internal oxidation. How is the phase composition (Cu2O, Al2O3) determined?

Why don't the authors express temperature in degrees Celsius or Kelvin? Why does copper oxidation occur to Cu2O and not CuO. How do the authors explain this? How the phase (Cu2O) was determined?

Author Response

Reviewer-2

Review of the article.

  1. The article expresses the reasons for the lack of recrystallization of the Cu-Al.The results are interesting, but several questions arise.

Ans: We thank the reviewer for his comments.

  1. At what time and at what oxygen pressure does the internal oxidation of the alloy components and the formation of Al2O3, Cu2O.

The amount of Al2O3 is controlled by the Al (0.15 wt%) and O content (0.17 wt%) in the alloy powder after the internal oxidation.  In this case, the log of oxygen partial pressure is approximately log (PO2) = -6.0.

  1. What conditions need to be created to prevent internal oxidation.How is the phase composition (Cu2O, Al2O3) determined?

In this case, the internal oxidation has been carried to get fine dispersion of Al2O3 in Cu matrix. Without the internal oxidation, it would be just a Cu-0.15 wt. % Al dilute alloy. The level of Al and O in the alloys determines the volume fraction of Al2O3. In the as received condition, the volume fraction of Al2O3 is ~1%, as specified by the company. We use HRTEM to determine the structure of these oxide phases.

  1. Why don't the authors express temperature in degrees Celsius or Kelvin?Why does copper oxidation occur to Cu2O and not CuO. How do the authors explain this? How the phase (Cu2O) was determined?

Ans: We used Celsius throughout the text.  The Cu2O is the stable phase along with Cu below the eutectic temperature in the range of 0.1 to 11 wt. % O. The oxygen content in the alloy is 0.17 wt.%, and the oxidation has been carried at a lower partial pressure of O2 to ensure CuO does not form. We have observed nanocrystalline Cu2O as determined by the HRTEM studies.   It forms epitaxially with Cu (see Fig. 10).   

Reviewer 3 Report

In this manuscript, the authors investigated the mitigated recrystallization process of the cold worked Cu-Al2O3 nanocomposite by transmission electron microscopy. The results illustrate that the cold worked Cu-Al2O3 composite did not recrystallize up to a temperature of 0.83Tm of Cu, which can be attribute the second phase Cu2O nanoparticles enhanced Zener drag, limiting the migration of grain boundary and the nucleation of defect free regions of different orientations. The results are interesting, and give some insights to understand the microstructure and recrystallization process of deformed metallic-oxide nanocomposite materials. After a careful review on the manuscript, I recommend the manuscript to publish in Nanomaterials after a revision.

(1)   Except for the Cu peaks in the XRD pattern in Figure 1b, there is an extra peak at ~ 35o, which should be confirmed and indexed in the Figure.

(2)   The angle of low-angle grain boundary should be measured and given in the revised manuscript.

(3)   In Figures 10-11, the IFFT images showing nanocrystalline Cu2O particles indicated by circles in the Figures 10b-11b do not match well with the contrast in the corresponding TEM images in Figures 10a-11a. It should be double checked

(4)   There are some typos, such as “γ-Al2O3” should be “γ-Al2O3” in Figure caption of Figure 3a, which should be checked.

(5)   The recent papers related to the crystallization of nanoparticles (like Chemical Reviews, 2022, 122, 23, 16911-16982; Nanoscale Horizons, 2019, 4, 1302-1309; Advanced Science, 2018, 6, 1802131) should be cited.

 Minor editing of English language required

Author Response

In this manuscript, the authors investigated the mitigated recrystallization process of the cold worked Cu-Al2O3 nanocomposite by transmission electron microscopy. The results illustrate that the cold worked Cu-Al2O3 composite did not recrystallize up to a temperature of 0.83Tm of Cu, which can be attribute the second phase Cu2O nanoparticles enhanced Zener drag, limiting the migration of grain boundary and the nucleation of defect free regions of different orientations. The results are interesting, and give some insights to understand the microstructure and recrystallization process of deformed metallic-oxide nanocomposite materials. After a careful review on the manuscript, I recommend the manuscript to publish in Nanomaterials after a revision.

Ans: We thank the reviewer for his useful comments.

  • Except for the Cu peaks in the XRD pattern in Figure 1b, there is an extra peak at ~ 35o, which should be confirmed and indexed in the Figure.

Ans: We have indexed the peak it is actually 111 of Cu2O.

  • The angle of low-angle grain boundary should be measured and given in the revised manuscript.

Ans: The angle is approximately 10⁰.

  • In Figures 10-11, the IFFT images showing nanocrystalline Cu2O particles indicated by circles in the Figures 10b-11b do not match well with the contrast in the corresponding TEM images in Figures 10a-11a. It should be double checked.

The IFFT image magnification is different from the image, which might be a reason that it might appear different.

(4)   There are some typos, such as “γ-Al2O3” should be “γ-Al2O3” in Figure caption of Figure 3a, which should be checked.

We have modified the typos in the figure caption.

(5)   The recent papers related to the crystallization of nanoparticles (like Chemical Reviews, 2022, 122, 23, 16911-16982; Nanoscale Horizons, 2019, 4, 1302-1309; Advanced Science, 2018, 6, 1802131) should be cited.

Ans:  We thank the reviewer for those references. However, they are not directly related. Here the term recrystallization is associated with the formation of defect free region. The recrystallization usually takes place when a deformed metal or metallic alloy heated to elevated temperature before grain growth. The driving force is the stored energy. It is not the nucleation in the classical nucleation sense.

Reviewer 4 Report

The comments are follows.

Oxide content in Cu is normally kept at extra low level to avoid hydrogen disease. Authors alloyed by Oxygen to increase  high temperature strength. At a high temperature the problems of Oxygen can become more complex. Was it considered in the present study? 

Zener pinning effect should be estimated considering both fraction and size of precipitates, thus, d/f ratio, small fraction if the size is small can be also effective. It should be considered when authors discuss an effect of alumina in the introduction.

The last and last to the end  paragraphs of the introduction is a part of conclusion of the present research. Consider revising it to include only background and purpose of the study.

Corrected the deformation level to a strain reduction. 

Explain how the rolling was processed.

What was a reason of a rapid cooling from 900C? 

Authors repeated several time that the fraction was 1%. How it was estimated?

Add SAED to Fig1a.

The meaning of the first sentence in the second paragraph page 9 is unclear. What are clusters? Where they are in the micrographs? How oxygen dissolves oxygen?

Twinning is discussed , show twins in micrographs. 

Some minor changes are required, example in the comment above,please carefully read the manuscript or improve readability. 

Author Response

Reviewer-4

The comments are follows.

  1. Oxide content in Cu is normally kept at extra low level to avoid hydrogen disease. Authors alloyed by Oxygen to increase high temperature strength. At a high temperature the problems of Oxygen can become more complex. Was it considered in the present study? 

Ans: We agree with the reviewer that these alloys are prone to hydrogen embrittlement at elevated temperatures. However, it happens in reducing atmospheres. The internal oxidation has been done to get fine dispersion of Al2O3 for high temperature strength in oxidizing atmosphere. At high temperature, in addition Al2O3, the nanocrystalline Cu2O forms as a fine dispersion in the matrix.  A small amount of boron at 250 ppm level can be added to eliminate hydrogen embrittlement [see ref. 10].  

  1. Zener pinning effect should be estimated considering both fraction and size of precipitates, thus, d/f ratio, small fraction if the size is small can be also effective. It should be considered when authors discuss an effect of alumina in the introduction.

Thanks for this comment. Yes, we have used both the fraction and the size for estimating the Zener pinning effect. We modified the introduction.

  1. The last and last to the end paragraphs of the introduction is a part of conclusion of the present research. Consider revising it to include only background and purpose of the study.

Thanks. We have removed the last part of the introduction to avoid confusion.

  1. Corrected the deformation level to a strain reduction. 

     Ans: We have modified it in the revised version.

  1. Explain how the rolling was processed.

 Ans: The rolling has been carried out at room temperature to get plates of 3 mm in thickness. The strain level is 80%.    

  1. What was a reason of a rapid cooling from 900C? 

We wanted to preserve the microstructure by cooling in air.  We have modified the text.

  1. Authors repeated several time that the fraction was 1%. How it was estimated? Add SAED to Fig1a.

This is specified by the company. It is controlled by the Al (0.15 wt%) and oxygen (0.17 wt%) content in the alloy powder. The amount (volume fraction) of Al2O3 at this oxygen level is around 1%. We have modified the text.

  1. The meaning of the first sentence in the second paragraph page 9 is unclear. What are clusters? Where they are in the micrographs? How oxygen dissolves oxygen?

We have modified the text in this version.

  1. Twinning is discussed, show twins in micrographs. 

We have shown micro faults/twins in the micrograph. See Figs. 9 (a-c). Results and discussion are given in page 8 and page 13, respectively.

Comments on the Quality of English Language

Some minor changes are required, example in the comment above, please carefully read the manuscript or improve readability. 

We have modified the text.

Round 2

Reviewer 2 Report

My comments have been taken into account.

Reviewer 4 Report

Authors